# Artificial Intelligence-Assisted Processing of Anterior Segment OCT Images in the Diagnosis of Vitreoretinal Lymphoma

**DOI:** 10.3390/diagnostics13142451

**Published:** 2023-07-23

**Authors:** Fabrizio Gozzi, Marco Bertolini, Pietro Gentile, Laura Verzellesi, Valeria Trojani, Luca De Simone, Elena Bolletta, Valentina Mastrofilippo, Enrico Farnetti, Davide Nicoli, Stefania Croci, Lucia Belloni, Alessandro Zerbini, Chantal Adani, Michele De Maria, Areti Kosmarikou, Marco Vecchi, Alessandro Invernizzi, Fiorella Ilariucci, Magda Zanelli, Mauro Iori, Luca Cimino

**Affiliations:** 1Ocular Immunology Unit, Azienda USL-IRCCS, 42123 Reggio Emilia, Italy; fabrizio.gozzi@ausl.re.it (F.G.); pietro.gentile@unimore.it (P.G.); luca.desimone@ausl.re.it (L.D.S.); elena.bolletta@ausl.re.it (E.B.); valentina.mastrofilippo@ausl.re.it (V.M.); chantal.adani@ausl.re.it (C.A.); 2Medical Physics Unit, Azienda USL-IRCCS, 42123 Reggio Emilia, Italy; marco.bertolini@ausl.re.it (M.B.); laura.verzellesi@ausl.re.it (L.V.); valeria.trojani@ausl.re.it (V.T.); mauro.iori@ausl.re.it (M.I.); 3Clinical and Experimental Medicine Ph.D. Program, University of Modena and Reggio Emilia, 41125 Modena, Italy; 4Molecular Pathology Unit, Azienda USL-IRCCS, 42123 Reggio Emilia, Italy; enrico.farnetti@ausl.re.it (E.F.); davide.nicoli@ausl.re.it (D.N.); 5Clinical Immunology, Allergy and Advanced Biotechnologies Unit, Azienda USL-IRCCS, 42123 Reggio Emilia, Italy; stefania.croci@ausl.re.it (S.C.); lucia.belloni@ausl.re.it (L.B.); alessandro.zerbini@ausl.re.it (A.Z.); 6Ophthalmology Unit, Azienda USL-IRCCS, 42123 Reggio Emilia, Italy; michele.demaria@ausl.re.it (M.D.M.); areti.kosmarikou@ausl.re.it (A.K.); marco.vecchi@ausl.re.it (M.V.); 7Eye Clinic, Luigi Sacco Hospital, Department of Biomedical and Clinical Science, University of Milan, 20157 Milan, Italy; alessandro.invernizzi@gmail.com; 8Faculty of Health and Medicine, Save Sight Institute, University of Sydney, Sydney, NSW 2000, Australia; 9Hematology Unit, Azienda USL-IRCCS, 42123 Reggio Emilia, Italy; fiorella.ilariucci@ausl.re.it; 10Surgical Oncology Unit, Azienda USL-IRCCS di Reggio Emilia, 42123 Reggio Emilia, Italy; magda.zanelli@ausl.re.it; 11Department of Surgery, Medicine, Dentistry and Morphological Sciences, with Interest in Transplants, Oncology and Regenerative Medicine, University of Modena and Reggio Emilia, 41124 Modena, Italy

**Keywords:** anterior segment optical coherence tomography, AS-OCT, vitreoretinal lymphoma, VRL, radiomic features, machine learning, AI, artificial intelligence

## Abstract

Anterior segment optical coherence tomography (AS-OCT) allows the explore not only the anterior chamber but also the front part of the vitreous cavity. Our cross-sectional single-centre study investigated whether AS-OCT can distinguish between vitreous involvement due to vitreoretinal lymphoma (VRL) and vitritis in uveitis. We studied AS-OCT images from 28 patients (11 with biopsy-proven VRL and 17 with differential diagnosis uveitis) using publicly available radiomics software written in MATLAB. Patients were divided into two balanced groups: training and testing. Overall, 3260/3705 (88%) AS-OCT images met our defined quality criteria, making them eligible for analysis. We studied five different sets of grey-level samplings (16, 32, 64, 128, and 256 levels), finding that 128 grey levels performed the best. We selected the five most effective radiomic features ranked by the ability to predict the class (VRL or uveitis). We built a classification model using the xgboost python function; through our model, 87% of eyes were correctly diagnosed as VRL or uveitis, regardless of exam technique or lens status. Areas under the receiver operating characteristic curves (AUC) in the 128 grey-level model were 0.95 [CI 0.94, 0.96] and 0.84 for training and testing datasets, respectively. This preliminary retrospective study highlights how AS-OCT can support ophthalmologists when there is clinical suspicion of VRL.

## 1. Introduction

Vitreoretinal lymphoma (VRL) is the most common uveitis masquerade syndrome and thus a diagnostic challenge [1]. The insidious clinical presentation as chronic and recurrent uveitis, combined with the transient response to lymphocytolytic steroids, leads to diagnostic and therapeutic delay, often resulting in a poor prognosis with high mortality if the central nervous system (CNS) is involved [2]. Therefore, a timely and accurate diagnosis is essential to improve the prognosis of patients with VRL, and overcoming the difficulties encountered in its diagnosis is an urgent need [3].

Definite diagnosis of VRL is based on different laboratory methods that analyse the vitreous and/or aqueous sample. The gold standard of cytology combined with immunohistochemistry shows the presence of large-sized atypical cells with irregular hyperchromatic nuclei, recognisable nucleolus, and small cytoplasm, which are characteristically CD20+, diffuse large B-cell lymphomas (DLBCLs) being present in over 95% of cases of VRL [4]. Scanty amounts of vitreous samples, rare lymphoma cells often admixed with inflammatory cells, and inadequate cellular preservation are the main confounding factors, which make diagnosis particularly difficult, thus causing delay [5]. Although not generally diagnostic in isolation, ancillary tests are therefore used to support the diagnosis. These tests include flow cytometry, detection of interleukins 6 (IL-6) and 10 (IL-10) and their ratio IL10:IL6, clonality analyses through immunoglobulin heavy (IgH) chain rearrangements, and MYD88 mutation analysis [5,6,7]. The diagnostic laboratory tools for VRL ideally are tests with high positive predictive value, thereby being able to detect as many cases of VRL as possible while carefully distributing the small sample. Currently, however, no one method or combination of tests allows a diagnosis in every case of VRL. Next-generation sequencing (NGS) analysis represents a recent technology that offers high performance, making it possible to find particular genomic alterations. Although there are few NGS analysis studies in VRL, this technique is catching on as a useful additional diagnostic method using even the least quantities of vitreous humour [8].

In this context, vitreoretinal multimodal imaging could aid diagnosis when VRL is clinically suspected [9]. Fluorescein angiography shows well-defined hypofluorescent lesions corresponding to the lymphomatous infiltrates in the early and late phases. Other fluorangiographic findings in VRL include punctate hyperfluorescent window defects, optic disc leakage, and patchy late hyperfluorescence [10]. Fundus autofluorescence shows a granular hyperautofluorescence pattern [11]. The literature amply describes the well-known use of optical coherence tomography (OCT) to recognise subretinal pigment epithelium (sub-RPE) and subretinal infiltrates [12]. In fact, hyper-reflective nodular lesions within or below the RPE are considered a highly suggestive finding in VRL but should be carefully differentiated from drusen or choroidal neovascularisation [13]. Currently, posterior spectral domain OCT represents the only helpful diagnostic OCT in suspected VRL [14].

Thanks to the improvement in imaging technology, swept-source anterior segment OCT (AS-OCT) has been proven useful to identify cells and indirectly quantify flare in the anterior chamber of patients with uveitis [15,16]. Likewise, the posterior segment OCT image has such high resolution that it correlates with the clinical grading of vitreous haze [17,18,19,20]. Zicarelli et al. were able to obtain OCT images of the anterior vitreous in the area just behind the lens, which had previously been inaccessible to this imaging technique due to its location, too forward for the posterior segment and too back for anterior segment OCT [21]. These authors pushed the device towards the eye being studied and disabled the tracking, thus not following the recommendations on its use. As a result, the cornea was flipped onto the lens, which was in the upper part of the image, and the structures behind it could be seen in the lower part of the scan. This revised capture technique produced a reflection artefact due to the projection of the cornea as a hyper-reflective curved band on the lens; it is important that the device is not pushed too far in order not to compromise the view of the vitreous chamber.

A key feature of VRL is the vitreous involvement represented by a characteristic infiltration due to the invasion of lymphoma cells that form sheets or strands with vitreous turbidity [6]. This is evident during the ophthalmological examination when carefully looking at the anterior vitreous. However, because this feature is difficult to recognise, it enters the differential diagnosis with the uveitis that causes significant vitreous haze; among these, the most frequent and confusing are Fuchs uveitis, sarcoidosis uveitis, and Behçet uveitis. The clinical picture cannot always, anyway, be matched to a defined nosological entity even after an extensive diagnostic uveitis work-up that includes brain magnetic resonance imaging, chest computed tomography, and invasive surgical procedures such as aqueous tap or diagnostic vitrectomy; these intraocular inflammations are called uveitis of unknown origin [22]. Fuchs uveitis is the only viral anterior uveitis to have significant vitreous involvement, especially anterior although not exclusively [23]. It should also be highlighted that Fuchs uveitis is bilateral albeit in few patients (5–10%), entering even more into the differential diagnosis with VRL in these cases [24]. Sarcoidosis uveitis is a bilateral granulomatous uveitis which can have intense vitritis, able to present as intermediate uveitis [25]. Behçet uveitis is a recurrent non-granulomatous uveitis associated with haemorrhagic or occlusive retinal vasculitis, and diffuse vitritis is a constant feature [26]. Moreover, to complicate the differential diagnosis process, anterior chamber involvement is also possible in VRL [27]. Particular types of vitritis have been associated with specific clinical entities such as vitritis in Fuchs uveitis and multiple whitish puff balls floating in the vitreous chamber in endogenous candida endophthalmitis [28,29]. As such, the possibility of imaging and objectively evaluating the anterior vitreous with AS-OCT would be a significant added value for the differentiation between vitreous haze, make it possible to detect a particular potential pattern in VRL, and could be an important ancillary test in its diagnosis.

Radiomics is a novel methodology in precision medicine that employs well-defined mathematical formulas to describe medical images quantitatively [30]. This approach is an application of artificial intelligence (AI) that calculates features directly from the images’ pixel values or filtered versions of the original captures. The mathematical formulas express the distribution and relationships between pixels and voxels within a specific region of interest (ROI) in the images. Haralick et al. showed how emphasising the relationships between the grey levels and the textural patterns could be used to detect different regions in an image (i.e., segmentation tasks) [31]. Later, textural information was applied in medical imaging [32,33,34]. Within a biomedical image, there are complex details that may be beyond the human eye’s perception; this is what radiomics is based on. Radiomic features are numerical descriptors that enable a non-invasive exploration of potential correlations between the information derived from the images acquired during the routine clinical pathway of the patient and the clinical or biological features seen on the images [35].

To our knowledge, no other study has investigated the ability of radiomic features to perform a differential diagnosis of the vitreous using OCT imaging. We hypothesised that there are particular radiomic features that make it possible to distinguish between VRL and vitritis.

## 2. Materials and Methods

We analysed patients evaluated at the Ocular Immunology Unit of the AUSL-IRCCS of Reggio Emilia (Italy) between January 2019 and December 2022; we considered biopsy-proven VRL and uveitis characterised by anterior vitritis, such as Fuchs uveitis, sarcoidosis uveitis, Behçet uveitis and uveitis of unknown origin, in differential diagnosis with VRL. We name the latter group “vitritis”, which will be used in the remainder of this article.

### 2.1. Image Acquisition

The images were obtained with swept-source AS-OCT ANTERION (Heidelberg Engineering, Heidelberg, Germany), which offers an improved signal penetration (wavelength 1300 nm), a very high resolution (axial and transverse resolution in tissue <10 μm and 30 μm, respectively) and an enhanced depth of scan (14 ± 0.5 mm) [21]. Before May 2022, the acquisitions were performed using the recommended acquisition settings of the OCT device (referred to here on as “old”). Conversely, the new acquisition technique described above will be referred to as “new” [21].

### 2.2. Image Segmentation

We extracted anonymised Digital Imaging and Communications in Medicine (DICOM) images directly from the OCT system. Next, two experienced ophthalmologists manually segmented the vitreous area in the images from the sub-lens region up to the end of the visible vitreous using a rectangular-shaped ROI as shown in Figure 1. The area of these ROIs depended on several variables, particularly the exam technique, the anatomical localisation of the slice, and the patient’s individual clinical conditions. We performed exams with the old and the new method of pushing the device forward, the latter one for patients referred after May 2022. Image width depended on the relative distance from the optical axis, and the length relied on the exam technique. If the selected area was less than 13,500 pixels, we did not include its respective images in our analysis according to similar results found for the radiomics predictive model applied in oncology [36]. In addition, we rejected slices with artefacts caused by the hyper-reflective bend zone generated by the cornea shadow on the anterior vitreous cavity, as mentioned above.

### 2.3. Radiomic Analysis

We used the Radiomics tool (Version 1.2.0.0 by Martin Vallières, publicly available at https://it.mathworks.com/matlabcentral/fileexchange/51948-radiomics, accessed on 5 July 2023) to extract radiomic features with the aforementioned image segmentations as inputs [37]. Radiomics software is a free program running on MATLAB^®^ (Mathworks, Natick, MA, USA). We used MATLAB version R2021b. This software was designed to calculate 43 features: three from the image histogram (variance, skewness, and kurtosis), nine from the Gray-Level Co-Occurrence Matrix (GLCM), 13 from the Gray-Level Run-Length Matrix (GLRLM), 13 from the Gray-Level Size Zone Matrix (GLSZM) and five from the Neighborhood Gray-Tone Difference Matrix (NGTDM) [37,38,39].

Before feature extraction from 2D images, we used the following pre-processing parameters: the number of grey levels (Ng = 16, 32, 64, 128, 256), equal quantisation (i.e., the ROI was equalised to increase the contrast), isotropic pixel size (8.843 µm), and ‘scanType’ equal to ‘Other’; no wavelet band-pass filtering was performed. The patients were preliminarily divided into two groups: training and testing. The first set was composed of 14 patients (seven vitritis—11 eyes—53% images of the training dataset, and seven VRLs—11 eyes—47% images), as well as the second one (10 vitritis—17 eyes—68% images of the testing dataset, and four VRLs—eight eyes—32% images). The dataset had a good balance in the training group in terms of acquisition method, lens type, and outcome.

The first step in the analysis consisted of feature scaling using Scikit learn [40].

### 2.4. Statistical Analysis

The correlation between the variables was studied using the Pearson correlation coefficient (r) for radiomic models built with five different sets of grey-level sampling (16, 32, 64, 128, and 256 levels).

Within the training dataset, five features were selected per each model using the xgboost python function; the features were selected using feature importance greater than 0.03. Then five models were built using the xgboost algorithm using early stopping round equal to 100 and a learning rate equal to 0.01. The areas under the receiver operating characteristic curves (AUCs) were calculated with their confidence interval for the training dataset using bootstrapping methods (500 repetitions were performed). In addition, the accuracy and precision were calculated for training and testing sets.

The models assigned to each image have the probability of belonging to the VRL class. If this probability was less than 0.5, the image was classified as belonging to the vitritis group, as the outcome was binary. Each eye was assigned to a group based on whether the majority of that eye’s images were classified as VRL or as vitritis.

Clinical features considered in the statistical analysis were age at diagnosis (years), sex, laterality, lens status (phakic or pseudophakic), and acquisition method (old and/or new).

The study was conducted in agreement with the principles of the Declaration of Helsinki and received approval from the local ethics committee (protocol n. 2019/0085664 Comitato Etico dell’Area Vasta Emilia Nord, Italy).

## 3. Results

Eleven patients with biopsy-proven VRL (a total of 19 eyes) and 17 patients (a total of 28 eyes) with vitritis were included in this retrospective study. Specifically, five patients with Fuchs uveitis, four patients with sarcoidosis uveitis, two patients with Behçet uveitis, and six patients with uveitis of uncertain origin were enrolled in the group of vitritis. Clinical features are summarised in Table 1. In both groups, there were more bilateral than unilateral cases, and unilateral uveitis were Fuchs uveitis or uveitis of uncertain origin. A total of 31 eyes were phakic and 16 were pseudophakic. There were 13 males and 15 females. The mean age of vitritis patients was 55 years old (20–79), while that of VRL patients was 72 (51–94) years old.

Overall, 3260/3705 (88%) AS-OCT images met our defined quality criteria, making them eligible for analysis. Specifically, 2131 images were from eyes of patients with uveitis, while 1129 were from eyes of patients with biopsy-proven VRL.

Table 2 shows the patient stratification per eye site, acquisition method, lens type, and sex.

The Pearson correlation coefficients between the variables for the five models are shown in the Appendix A. Age, sex, lens type (phakic or pseudophakic) and eye site (left or right) did not demonstrate any correlation with the outcome (r = 0.4, r = −0.1, r = 0.1, and r = 0.0, respectively). We observed a moderate correlation (r = −0.5) between vitritis outcome and acquisition method that was intrinsically dependent on the ROI area.

Table 3 reports the models’ results for the different sets of grey-level sampling considered in the radiomic analysis pre-processing. Model outcomes by each patient are available in the Appendix A. The model using 128 grey levels (Dataset Ng = 128) demonstrated the best performance in terms of AUC in both the training and the testing dataset, 0.95 [CI 0.94, 0.96] and 0.84, respectively; in this model, accuracy was 0.860 and 0.781 in the training and the testing dataset, respectively, and precision was 0.985 and 0.582 in the training and the testing dataset, respectively. Using this model, 41/47 (87%) of eyes corresponding to 23/28 (82%) of patients were correctly diagnosed as VRL or uveitis (see Appendix A), regardless of the exam technique used or lens status. We clarify that we have considered a patient as correctly classified only if both eyes were properly identified in case of bilateral involvement. The ROC curves relative to models considered in our study are shown in the Appendix A.

Table 4 gathers the Pearson correlation coefficients among the radiomic features employed in the 128 grey-level model since the latter is the one that proved to perform the best.

## 4. Discussion

The incidence of VRL has increased over the last decades thanks to improvements in diagnostics and advances in health care, resulting in longer life expectancies [41]. It must be emphasised that patients later found to be affected by VRL are being referred more frequently to a uveitis specialist because these tumours may mimic inflammatory eye disease [42]. For this reason, recognising VRL, especially, in differential diagnosis with uveitis is a great challenge; the high rate of diagnostic delay worsens not only the visual prognosis, it also, and more importantly, shortens the patient’s life [2]. It is important to objectively discriminate between vitreous infiltration in VRL and vitreous inflammation in uveitis. Indeed, the former has characteristics that are so beyond clinical uveitis grading that it can be defined as “muddy”, corresponding to aurora borealis and string of pearls in ultrawide-field imaging [43]. Vitreous morphological characteristics are traceable to two different etiopathogenic pathways: inflammation in the case of uveitis and cancer in the case of VRL [44,45].

Multimodal imaging could aid in detecting “muddy” vitreous. The only imaging valuable for diagnosing VRL is currently that of the posterior segment [9,12]. This study evaluated the possible usefulness of the AS-OCT in diagnosing this rare but dangerous tumour. To obtain this result, we based our predictive model on biopsy-proven lymphomatous cases to distinguish between vitreous involvement in VRL and vitritis. To verify the accuracy of AS-OCT, data from 28 patients (11 with biopsy-proven VRL and 17 with differential diagnosis uveitis) were used to train and test our model’s performance.

Our results show that age did not influence the model outcome. On the one hand, it is true that the vitreous undergoes morphological changes with age, leading to an increase in opacities perceived as floaters, which sometimes functionally disturb vision (the so-called “vision degrading myodesopsia”) [46,47]. On the other hand, it has recently been highlighted that the structural and morphological alterations of the anterior vitreous can be analysed with OCT, which is able to distinguish between the vitreous cisterns and lacunae, which are hyporeflective, and the hyper-reflective macromolecular aggregates of collagen that form the bundles of linear fibrils typical of the aging vitreous [48,49].

This is the first study that evaluates radiomics in uveitis. However, in ophthalmology, radiomic models have been preliminarily studied in diabetic retinopathy, retinal venous occlusions, and relative response to anti-vascular endothelial growth factors (VEGF) [50,51]. The basis of radiomics consists of extracting well-defined, hand-crafted features from the image pixels in order to grasp what is not explainable using the clinician’s eye [30,52]. Indeed, looking at the OCT scans of the anterior vitreous, clinicians would not be able to distinguish between vitreous involvement due to VRL or vitritis. On the contrary, radiomics performs a mathematical analysis on each single OCT capture, so it is able to catch differences in the pattern of all analysed images and allow it to reach the correct diagnosis in 87% of eyes and in 82% of patients. Our study shows that radiomic processing of the AS-OCT images could be an important supportive test helping the clinician in the diagnostic pathway of VRL.

In other fields of medicine, especially in oncology, radiomic analyses using statistical learning methods by analysing ultrasound [53,54], computed tomography [55,56,57], magnetic resonance [58,59], or positron emission tomography [60] images are very useful for differentiating between cancer and inflammation [61].

It is important to note that we did not use the acquisition method as a feature to train the models because it would have been a bias, given the retrospective nature of this study. Due to the rarity of VRL, it is very difficult to collect a sizable cohort of these patients.

Based on the results listed in Table 4, we found that several features were collinear, which could cause a problem with the proposed radiomic model. However, the xgboost algorithm was not affected by collinearity issues because it consisted of a decision tree ensemble classifier [62].

It is worth noting the meaning of the radiomic features selected to build the discriminant model. GLRLM_LRHGE quantifies grey-level runs (i.e., the number of consecutive pixels having the same grey-level value). GLCM_Correlation shows the linear dependency of grey-level values on their respective pixels in the grey-level co-occurrence matrix (0 = perfect decorrelation, 1 = ideal correlation). NGTDM_Coarseness measures the average difference between the central pixel’s grey level and that of its neighbour; this is an indicator of the spatial rate of grey-level variation. A higher value means a lower spatial change rate and a locally more uniform texture. NGTDM_Strength measures the primitive shapes in the image (i.e., the presence of simple elements such as arcs, squares, or other simple shapes in the image). A higher value means the primitives are easily defined and visible; it also means a slow change in intensity but a larger coarse difference in grey-level tones. GLSZM_HGZE quantifies the distribution of the higher grey-level-connected pixels that share the same grey-level intensity. A higher value indicates a greater proportion of higher grey-level tones and size zones in the image. GLRLM_SRE measures the distribution of short run lengths. A higher value indicates shorter run lengths and more refined texture. NGTDM_complexity describes the presence of many primitive components in the image; it also measures the image’s non-uniformity and rapid changes in grey-level intensity [63].

Our research shows that using a higher number of grey levels can highlight finer variations in the image as well as the complexity of their arrangement in the vitreal chamber. On the other hand, too many grey levels may introduce a bias in the model prediction caused by image noise. Decreasing the number of levels is a sort of complexity reduction, meaning that their ability to detect different details in the vitreal cavity diminishes. A trade-off must be considered; our analysis found it in 128 grey levels. The selected features to build radiomic models have in common their ability to describe how the higher number of pixels (normal vitreous tends to be represented with zero values) are arranged, and their relative displacement is better described in models with higher grey levels.

There has recently been an increasing number of studies employing deep learning in differential diagnosis [64]. However, we decided to use a machine learning method because we did not have a sufficient number of patients/eyes available due to VRL being a rare pathology. The optimal starting point should be to adopt machine learning methods to assess the feasibility of using AI techniques for this unexplored kind of imaging. A future deep learning-based study is possible provided there are more data.

In summary, the strength of this work is the relative simplicity of application of the described method based on a machine learning technique that allows calculating a probability of discrimination of VRL in a short time. Moreover, this study has some limitations. First, the retrospective nature of the analysis limited the consistency of the data. Second, the small sample size decreased the power of our statistical analysis. Third, this was a single-centre study. Because of these limitations, we could not perform a harmonisation of our data because it was not possible to apply the methods described in the literature [65,66,67,68].

## 5. Conclusions

We built a classification model using the xgboost python function; through our model, 87% of eyes were correctly diagnosed as VRL or uveitis, regardless of the exam technique used or lens status.

This preliminary retrospective study highlights how the AS-OCT can support the clinician in suspected VRL. Clearly, it should be emphasised that it is still necessary to collect a vitreous and/or aqueous sample to confirm a diagnosis, but AS-OCT is a quick and easy-to-use additional tool for deciding whether a diagnostic vitrectomy is required. This makes it possible to achieve the goal of reducing diagnostic delay, which would improve not only the patient’s visual prognosis but above all the *quoad vitam* prognosis. Further multicentre studies with a larger population sample are needed to confirm our results.

## Figures and Tables

**Figure 1 diagnostics-13-02451-f001:**
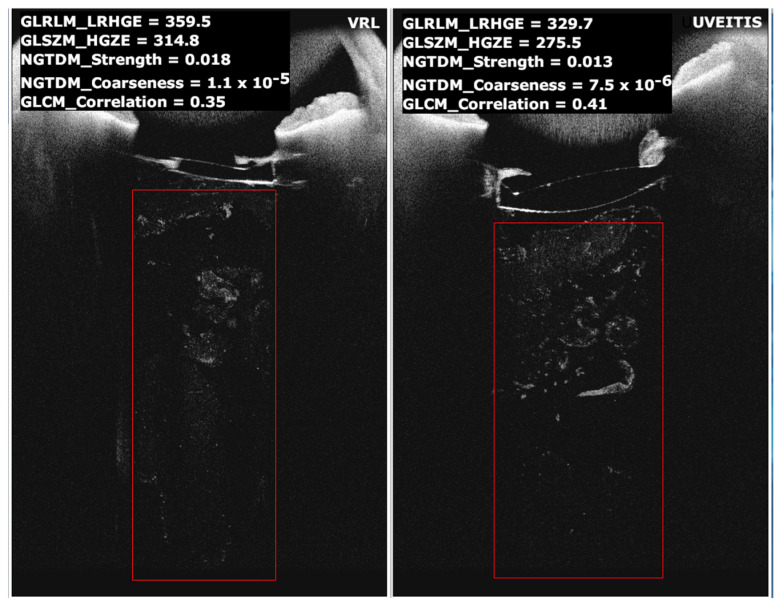
Comparison of two images: one patient with vitreous involvement in VRL (**left**) and one with vitritis (**right**). The rectangular-shaped ROI shows the area where the software calculated the radiomic features. In the upper left are shown, for example, the numerical value of the features used by the model built using 128 grey levels.

**Table 1 diagnostics-13-02451-t001:** Database representation showing the patient selection and images number used for training and testing groups. The lens description per each eye site is also reported (L: left or R: right). F: female; M: male; N.A.: not acquired; FU: Fuchs uveitis; OS: ocular sarcoidosis; BU: Behçet uveitis; UUO: uveitis of uncertain origin.

Patient Code	Patient Sex	Analyzed Images (n)	Label	Site	Age (y)	Acquisition Method	Training	Testing	Lens Type Left Eye (L)	Lens Type Right Eye (R)
433	F	124	Vitritis (FU)	R,L	26	both	X		Phakic	Phakic
434	M	98	Vitritis (OS)	R,L	57	new	X		Phakic	Phakic
439	F	132	Vitritis (OS)	R,L	74	new	X		Phakic	Phakic
440	M	99	Vitritis (FU)	L	50	new	X		Phakic	N.A.
444	M	97	Vitritis (FU)	L	39	new	X		Phakic	N.A.
445	M	114	Vitritis (OS)	R,L	52	new		X	Pseudo-phakic	Pseudo-phakic
405	M	111	Vitritis (FU)	R	38	old	X		N.A.	Phakic
435	F	96	Vitritis (BU)	R,L	20	new	X		Phakic	Phakic
437	F	239	Vitritis (BU)	R,L	26	new		X	Phakic	Phakic
438	M	95	Vitritis (OS)	R,L	46	new		X	Pseudo-phakic	Pseudo-phakic
448	F	217	Vitritis (FU)	R,L	65	old		X	Phakic	Phakic
410	F	109	Vitritis (UUO)	R,L	79	old		X	Pseudo-phakic	Pseudo-phakic
446	F	110	Vitritis (UUO)	L	66	both		X	Phakic	N.A.
466	F	90	Vitritis (UUO)	R	62	both		X	N.A.	Phakic
468	F	124	Vitritis (UUO)	R,L	78	both		X	Pseudo-phakic	Pseudo-phakic
491	M	174	Vitritis (UUO)	R,L	75	both		X	Pseudo-phakic	Pseudo-phakic
493	F	102	Vitritis (UUO)	L	79	both		X	Pseudo-phakic	N.A.
393	F	56	VRL	R,L	91	New	X		Pseudo-phakic	Pseudo-phakic
432	M	132	VRL	R,L	73	new		X	Phakic	Phakic
436	M	166	VRL	R,L	76	new	X		Pseudo-phakic	Phakic
442	F	62	VRL	R,L	88	new	X		Phakic	Pseudo-phakic
447	M	95	VRL	R	99	new	X		N.A.	Pseudo-phakic
103	M	82	VRL	R,L	55	old		X	Phakic	Phakic
173	M	145	VRL	R,L	51	old	X		Phakic	Phakic
186	M	18	VRL	L	58	old	X		Phakic	N.A.
363	F	80	VRL	R,L	71	old		X	Phakic	Phakic
364	F	32	VRL	L	82	old	X		N.A.	Phakic
398	F	261	VRL	R,L	58	old		X	Phakic	Phakic

**Table 2 diagnostics-13-02451-t002:** Patients’ stratification per eye site, acquisition method, lens type, and sex. *n*: number of images for each class; y: years; L: left; R: right; M: male; F: female; N.A.: not acquired.

	Site															
	L								R							
	Acquisition Method								Acquisition Method							
	New				Old				New				Old			
	Lens Type				Lens Type				Lens Type				Lens Type			
	Phakic		Pseudo-Phakic		Phakic		Pseudo-Phakic		Phakic		Pseudo-Phakic		Phakic		Pseudo-Phakic	
	Sex		Sex		Sex		Sex		Sex		Sex		Sex		Sex	
	M	F	M	F	M	F	M	F	M	F	M	F	M	F	M	F
Vitritis (n)	247	326	161	44	0	40	19	98	47	432	172	27	111	113	31	64
Average age (y)	47.1	46.3	63.8	78	NA	41.9	78	79	57	45.4	59	78	38	46.7	66	79
VRL (n)	44	37	94	0	129	191	23	39	137	0	95	25	116	182	0	17
Average age (y)	73	88	76	NA	53.2	61.3	76	91	74.1	NA	94	88	52.4	64.5	NA	91

**Table 3 diagnostics-13-02451-t003:** Radiomic model performance for the different grey levels (Ng). Accuracy, precision, and AUC are reported. The last column on the right describes the five features selected in each model.

	Accuracy (Train) (Test)	Precision (Train) (Test)	AUC (Train [CI 95%]) (Test)	Radiomic Features Selected
Dataset Ng = 16	0.878	0.968	0.947 [0.937–0.956]	GLCM_Homogeneity, NGTDM_Busyness, GLRLM_LRHGE, NGTDM_Coarseness, GLCM_Correlation
	0.735	0.525	0.813	
Dataset Ng = 32	0.844	0.987	0.938 [0.928–0.949]	GLRLM_LRHGE, GLCM_Contrast, NGTDM_Coarseness, GLCM_Correlation, GLCM_Homogeneity
	0.825	0.790	0.798	
Dataset Ng = 64	0.860	0.995	0.942 [0.932–0.951]	GLRLM_LRHGE, GLCM_Homogeneity, NGTDM_Coarseness, GLCM_Correlation, GLCM_Contrast
	0.827	0.798	0.809	
Dataset Ng = 128	0.860	0.985	0.949 [0.940–0.958]	GLRLM_LRHGE, NGTDM_Strength, GLSZM_HGZE, NGTDM_Coarseness, GLCM_Correlation
	0.830	0.795	0.843	
Dataset Ng = 256	0.853	0.990	0.945 [0.935–0.954]	GLRLM_LRHGE, NGTDM_Coarseness, GLCM_Correlation, GLRLM_SRE, NGTDM_Complexity
	0.785	0.589	0.841	

**Table 4 diagnostics-13-02451-t004:** Radiomic feature Pearson correlation coefficient used in the 128 grey-level model.

	GLRLM_LRHGE	NGTDM_Strength	GLSZM_HGZE	NGTDM_Coarseness	GLCM_Correlation
GLRLM_LRHGE	1.00				
NGTDM_Strength	0.54	1.00			
GLSZM_HGZE	0.43	0.81	1.00		
NGTDM_Coarseness	0.59	0.86	0.59	1.00	
GLCM_Correlation	0.44	0.44	0.48	0.37	1.00

## Data Availability

The data are contained within the article or in the Appendix A.

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
