# Peer review of "Artificial Intelligence-Assisted Processing of Anterior Segment OCT Images in the Diagnosis of Vitreoretinal Lymphoma"

_diagnostics, 2023, doi:10.3390/diagnostics13142451_

Round 1
Reviewer 1 Report
Thanks for the opportunity to review your work.
This is a detailed study looking at whether OCT images of the vitreous can be used to differentiate vitreoretinal lymphoma (VRL) from uveitis. The OCT used is amongst the best available, and there is a good number of VRL cases which is a rare condition.
When studying a diagnostic test the patient selection is crucial. Of course most patients with uveitis are young adults and most patients with VRL are elderly. Unfortunately this is a serious confounder as vitreous clearly changes with age, and the normal age-related clumping of vitreous fibres (syneresis) could account for the diagnostic performance of the OCT machine. I think this should be controlled using images of normal volunteers who are age matched to both uveitis and VRL groups.
I would suggest a statistical review, as I do not feel qualified to judge the merit of the analysis. The analysis is based on Pearson correlation coefficients (r) but this is perhaps not the best statistic when the outcomes of the model are binary, and when the variables in the model are not normally distributed nor have linear relationships. I think a logistic regression would be better.
I found the results very opaque. The supplementary materials are impossible to interpret. The AI models all seem to have a very high AUC, but I don't know that this is valid when there are only 14 patients in the training set. Each patient provides a lot of data points, but I don't know this makes the conclusions stronger. The variables included in each model are very hard to interpret. In the discussion the paragraph defining each part of the model is only a partial solution to this, as the model does not seem to allow us to interpret what aspects of an OCT image make it look more like VRL or more like uveitis.
I think the paper would benefit from some correlation with clinical findings. As the premise of the paper is that lymphoma has qualitatively different vitreous, described as things like 'muddy' or 'aurora' or 'pearls' it would be interesting to look at correlation between the grade of vitreous inflammation and specific clinical findings, and the various radiometric variables of the OCT analysis.
Reviewer 2 Report
This study attempted to diagnose ocular lymphoma using artificial intelligence technology.
1. Is it possible to distinguish other vitreous opacities, such as vitreous hemorrhage, from ocular lymphoma?
2. The paper “A Radiomics Model from Joint FDG-PET and MRI Texture Features for the Prediction of Lung Metastases in Soft-Tissue Sarcomas of the Extremities” is not publicly available. Therefore, I cannot check the codes that the authors used. Please reveal the code the authors used for this study.
3. Features seem intuitively difficult to understandable. Explain these features using the image as an example.
4. Image analysis using deep learning is not shown in the study at all. It is out of touch with recent research trends. Compare and analyze the results of deep learning research.
5. Please show the ROC curves.
6. Learning from a small number of patients is such a critical drawback. Lymphoma is a very rare disease. In this regard, consider the following research as a point of discussion. “Feasibility study to improve deep learning in OCT diagnosis of rare retinal diseases with few-shot classification”.
Round 2
Reviewer 1 Report
It seems I'm the only reviewer, and I am not sure that this is sufficient peer review for your work. I think the editors should seek a statistical review and potentially a VRL expert.
I appreciate the addition of several older patients with uveitis to better control for the possibility that age-related syneresis explains the differences. I am not convinced that the new references prove anything, but you now make the clearer statement that age and syneresis do not explain the differences between VRL and uveitis patients.
The conclusion, that 87% of eyes are correctly designated as VRL or uveitis, seems at odds with table S2 that indicates that at least 7 of the 27 patients were misclassified. This is more like 25-30% misclassification.
Regarding my request for more clinical correlation, you seem to discount the clinical descriptions as qualitative and therefore not comparable to the radiomic variables. I think the point of your study is to find variables that describe these qualitative descriptions. If the premise of the study is that VRL vitreous looks different to uveitis vitreous, then you are not using OCT radiomics to find 'invisible' features of VRL, indeed the OCT is using visible or near IR light to image the vitreous so you are simply using quantitative methods to descibe the appearance of the VRL vitreous.
Reviewer 2 Report
The manuscript has been improved.
Author Response
We thank the Reviewer for his/her valuable suggestions.
